# Thoracic Endovascular Aortic Repair for Aortobronchial Fistula 18 Years after Graft Replacement of the Descending Aorta

**Masato Hayakawa** [1,*]**, Takaaki Nagano** [2] **, Motomu Miyagi** [1]**, Ryo Ikemura** [1]**, Satoshi Yamashiro** [1]
**and Kiyoshi Iha** [1]

1    Department of Cardiovascular Surgery, Chubu Tokushukai Hospital, Kitanakagusuku 901-2393, Okinawa, Japan
2    Department of Thoracic and Cardiovascular Surgery, Graduate School of Medicine,
     University of the Ryukyus, Nishihara 903-0215, Okinawa, Japan
*    Correspondence: yunta_sp@mail.goo.ne.jp; Tel.: +81-98-932-1110

**Abstract:** A 77-year-old woman who had undergone graft replacement of the descending aorta 18 years prior presented to our hospital complaining of a cough with bloody sputum. She was diagnosed with aortobronchial fistula by computed tomography. Thoracic endovascular aortic repair was performed, and the patient was discharged from the hospital without any major complications. Postoperatively, bloody sputum disappeared, and computed tomography examination at 12 months postoperatively showed that the preoperative infiltrative shadows in the lung fields were reduced. In conclusion, thoracic endovascular aortic repair is an effective treatment for aortobronchial fistula.

**Keywords:** aortobronchial fistula; thoracic endovascular aortic repair; descending aorta

## 1. Introduction

Aortobronchial fistula (ABF), although rare, is an extremely serious condition with a high mortality rate, if not promptly corrected [1,2]. The causes of ABF include thoracic aortic aneurysm, advanced lung cancer, lung infection, and graft replacement of the aorta and thoracic endovascular aortic repair (TEVAR) for thoracic aortic aneurysm; open surgical repair has been the conventional treatment of choice [3]. However, open surgical repair is associated with high perioperative complications and mortality rates. In contrast, TEVAR for ABF has recently been reported to produce excellent results [4,5]. TEVAR for ABF is less invasive; nonetheless, recurrent aortobronchial fistula and stent graft infection have been reported as problems associated with this surgery [3,5]. In this study, we describe a case of TEVAR for ABF, which developed 18 years after graft replacement of the descending aorta, and present a review of the literature.

## 2. Case Presentation

A 77-year-old woman underwent graft replacement of the descending aorta for a descending aortic aneurysm 18 years prior. Postoperatively, aortic dissection was observed at the proximal and distal anastomoses of the graft, and the patient was regularly followed up with computed tomography (CT) in the outpatient department. At 3 months prior to admission, the patient developed a cough with bloody sputum, and due to the persistent symptoms, she was admitted for a thorough examination. On admission, a contrast-enhanced CT scan showed infiltrative shadows and gaseous images in the lung fields around the graft (Figure 1). Gallium scintigraphy also showed abnormal accumulation of gallium consistent with the infiltration shadows and gas images of the lung fields observed on contrast-enhanced CT (Figure 2). Based on these imaging findings, ABF was diagnosed. The patient was considered to be at high risk owing to her advanced age and post-graft replacement of the descending aorta; therefore, TEVAR was performed on the 11th day after admission.

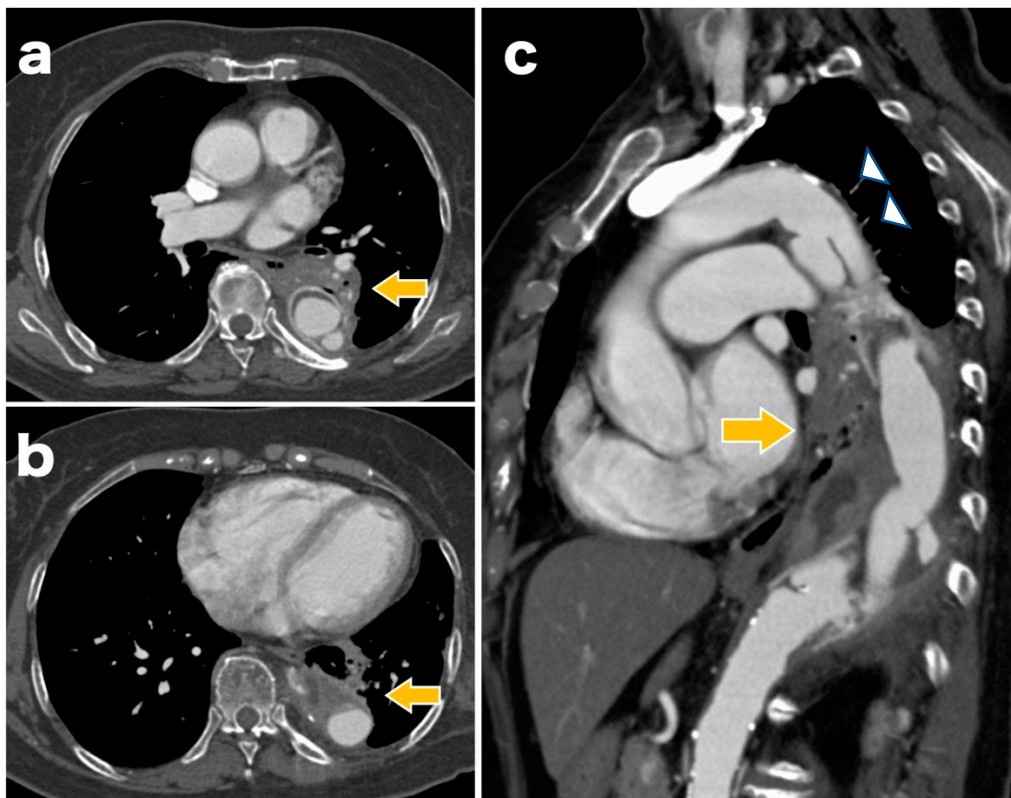

**Figure 1.** Contrast-enhanced computed tomography scan on admission showing infiltrative shadows and gaseous images in the lung fields around the graft (arrow). Although we could not directly signify the fistula between the aorta and the bronchus, the clinical history and CT findings led to the diagnosis of aortobronchial fistula ((**a**,**b**): axial images; (**c**): sagittal image). Chronic aortic dissection findings are also observed on the proximal side of the graft replacement (arrowhead). CT, computed tomography.

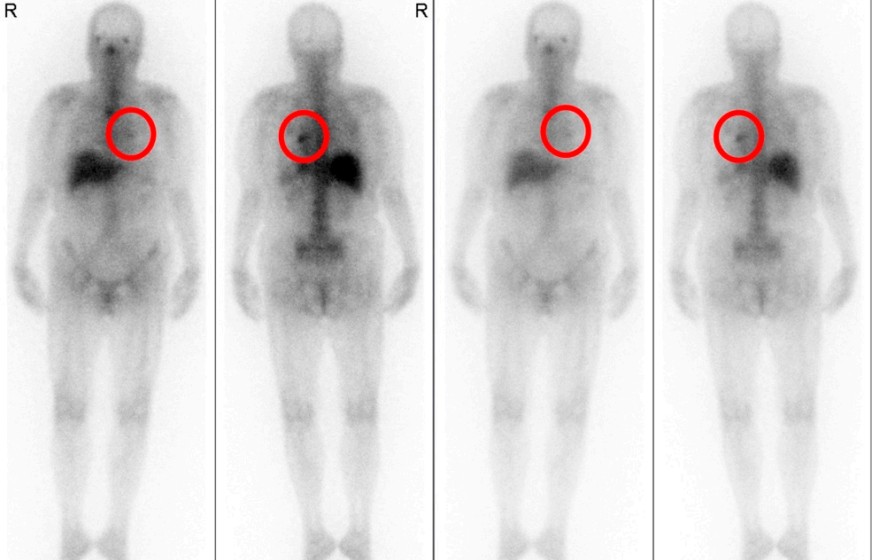

**Figure 2.** Gallium scintigraphy showing abnormal accumulation of gallium (red circle), consistent with the infiltrating shadow and gas image of the lung fields seen on contrast CT. CT, computed tomography.

The surgery was performed under general anesthesia. First, a 26 × 26 × 200 mm stent graft (GORE® TAG® Conformable Thoracic Stent Graft with ACTIVE CONTROL System, W.L. Gore and Associates, Flagstaff, AZ, USA) was inserted at the distal site via the left common femoral artery. Subsequently, a 28 × 28 × 150 mm stent graft was inserted at the proximal site, and the region from Zone 3 was implanted at the level of the 12th thoracic vertebra (Figure 3). The final aortograph showed a slight type 1a endoleak, but ballooning was not performed because of the known dissection findings on the proximal side.

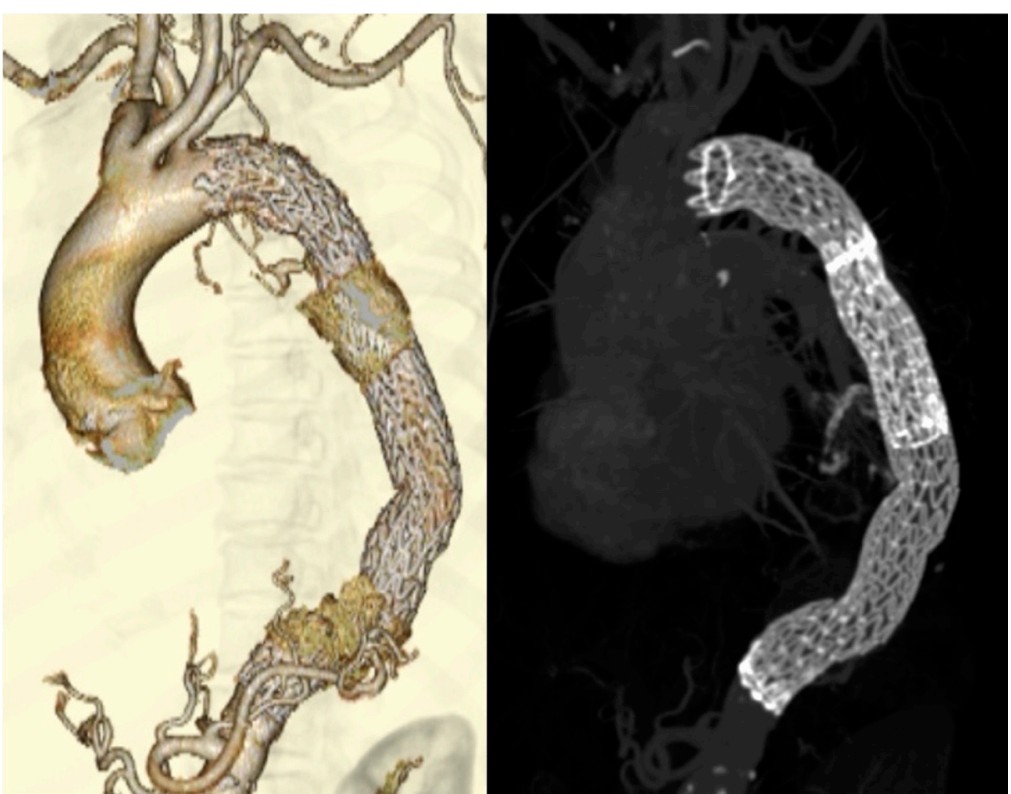

**Figure 3.** Postoperative contrast-enhanced computed tomographs showing no migration or fracture of the stent graft and no obvious endoleak.

Since the postoperative day 1, the amount of blood in the sputum began decreasing. A contrast-enhanced CT scan, performed on postoperative day 6, did not show the type 1a endoleak observed during the surgery. Fever with a body temperature ranging from 37.0 °C to 37.7 °C was observed until postoperative day 4, and no fever signs were noted thereafter. Sputum cultures were collected at 3 days before surgery, 2 days after surgery, and 16 days after surgery. Blood cultures were collected once on the day before surgery. However, the sputum culture showed no pathogenic bacteria and the blood culture was negative. Blood tests showed that, by postoperative day 6, the white blood cell (WBC) count had improved from 12,400/μL to 7200/μL, and C-reactive protein (CRP) had improved from 8.12 to 4.66 mg/dL. However, by postoperative day 16, the WBC count remained unchanged at approximately 7000/μL, but CRP increased to 7.89 mg/dL. Though the blood and sputum cultures obtained by postoperative day 16 were negative, the possibility of graft infection could not be ruled out. Therefore, levofloxacin (LVFX), a broad-spectrum drug effective against Gram-positive and Gram-negative bacteria, was administered orally (500 mg/day) since day 16 postoperatively. She was discharged on day 27 postoperatively, without fever and with improvement in CRP. The oral LVFX intake was continued for 14 days until postoperative day 30. At the first follow-up 7 days after discharge, the blood test showed that the WBC count was approximately 5000/μL, and the CRP level had improved to 0.52 mg/dL. The patient was followed up on an outpatient basis without

subsequent antibiotic administration, but no re-elevation of the inflammatory response was observed.

At 12 months postoperatively, a contrast-enhanced CT showed that the infiltrative shadows in the lung fields were reduced, and the endoleak had disappeared (Figure 4). No inflammation was observed, and blood in the sputum had disappeared, indicating a good postoperative course.

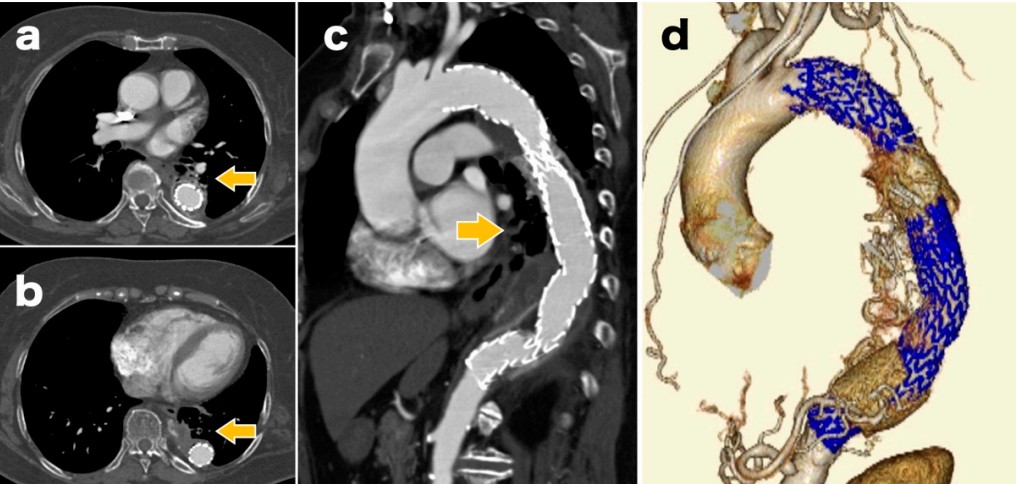

**Figure 4.** (**a**–**c**) Contrast-enhanced computed tomography at 12 months postoperatively shows infiltrative shadows in the lung fields around the graft and improvement of gaseous images (arrows). (**d**) No abnormality in the stent graft or endoleaks was observed.

## 3. Discussion

The causes of ABF include thoracic aortic aneurysm, advanced lung cancer, lung infection, and graft replacement or TEVAR for thoracic aortic aneurysms. Among these, previous thoracic aortic surgery is the most common cause, with 55% of all cases including a previous thoracic aortic surgery [3]. It has also been reported that the most frequent site of ABF is between the descending aorta, left lung, and/or bronchus [1,2]. Hemoptysis is a typical symptom, and shock may occur if the amount of bleeding is large [6]. A CT scan may aid in the prompt diagnosis of ABF with infiltrative shadows and hematoma images observed in the lung fields around the aorta. However, fistulas between the aorta and bronchus are rarely directly detectable on CT images [4]. In this case, the patient had hemoptysis, a cough that persisted for 3 months, and underwent descending aorta replacement surgery 18 years ago. In addition, CT images showed infiltrative shadows and gaseous images in the lung fields around the graft, suggesting a fistula between the graft and the left lung and/or bronchus, leading to ABF diagnosis.

Although open surgical graft replacement has been the treatment of choice for ABF, it has a high operative mortality rate, ranging from 15% to 41% [2]. TEVAR for ABF was first reported by Chuter et al. in 1996, and in recent years, several reports have suggested its efficacy, leading to its use as one of the main treatment options [7]. Piciche et al. studied 76 cases of ABF and reported a mortality rate of 6.6% in 15 post-TEVAR cases compared with 15.3% in 52 post-open surgical graft replacement cases [2]. Riesenman et al. reported a 30-day postoperative mortality rate of 1.5% in 67 patients treated with TEVAR for ABF [3], and Canaud et al. reported a 30-day mortality rate of 5.9% in 134 patients treated with TEVAR for ABF [5]. These reports show that TEVAR has better surgical outcomes than those of open surgical graft replacement. In our case, we chose TEVAR because the patient was elderly and considered high-risk after graft replacement of the descending aorta.

In contrast, ABF recurrence and stent graft infection are problems associated with TEVAR for ABF. Riesenman et al. reported ABF recurrence in 9% of patients during a mean follow-up period of 21.5 months [3], and Canaud et al. reported ABF recurrence in 11.1%

of patients during a mean follow-up period of 17.4 months, noting that the recurrence was attributed to residual tracheal defects after TEVAR [5]. There is also a risk of infection of the stent graft itself due to the residual fistula between the aorta and trachea after TEVAR. Therefore, if possible, surgical repairs, such as graft replacement and omental transfer flap, should be performed in the second stage [8]. The duration and type of antibiotic administration to prevent stent graft infection have not been clearly defined. A previous report suggested that intravenous antibiotics should be administered for at least 4 weeks followed by oral antibiotics, based on clinical findings and laboratory data [9]. However, it has been reported that some cases are cured by TEVAR alone and do not require prolonged antibiotic treatment [10].

In our case, the follow-up period was 15 months, and there was no ABF recurrence or signs of stent graft infection. Because of the prolonged elevated inflammatory response during hospitalization, LVFX was empirically administered for 14 days, and the postoperative blood and sputum cultures were negative. Therefore, other than LVFX, no other antibiotics were administered, and the patient was continuously monitored. However, as the antibiotic administration duration was short in this case, there is a possibility of ABF recurrence or stent graft infection in the future. Therefore, we will continue to closely follow up the patient with imaging studies, blood culture, and blood tests.

### 4. Conclusions

In this study, we report a case of ABF treated with TEVAR 18 years after graft replacement of the descending aorta. TEVAR is considered an effective treatment for ABF. In contrast, regular follow-up is necessary to avoid recurrence of ABF and graft infection.

**Author Contributions:** Conceptualization, M.H.; methodology, M.H.; validation, M.H., T.N., M.M., R.I., S.Y. and K.I.; formal analysis, M.H.; investigation, M.H.; resources, M.H.; data curation, M.H.; writing—original draft preparation, M.H.; writing—review and editing, M.H.; visualization, M.H.; supervision, T.N., R.I., S.Y. and K.I.; project administration, T.N., R.I., S.Y. and K.I. All authors have read and agreed to the published version of the manuscript.

**Funding:** This research received no external funding.

**Institutional Review Board Statement:** The ethical review and approval were waived for this study due to the description of a single clinical case.

**Informed Consent Statement:** Informed consent was obtained from the patient to publish this paper.

**Data Availability Statement:** The data used in this case report is available upon reasonable request from the corresponding author.

**Acknowledgments:** We would like to thank the patient for agreeing to have this case report published.

**Conflicts of Interest:** The authors declare no conflict of interest.

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
