# Peer review of "Thoracic Endovascular Aortic Repair for Aortobronchial Fistula 18 Years after Graft Replacement of the Descending Aorta"

_reports, doi:10.3390/reports5030034_

Round 1

Reviewer 1 Report

An interesting, educational and well written case report manuscript with an adequate review of the literature.  There some editing issues that the authors should consider and address.  The following are suggestions/comments regarding these editing issues.  Line 30, "...computed tomography (CT) in the outpatient department."  Lines 32 & 33, "On admission, a contrast-enhanced CT scan showed infiltrative....".  Line 69, "...after the surgery, a contrast-enhanced CT scan showed that the...".  Line 83, "...descending aorta, left lung, and/or bronchus [1,2]."  Lines 85 & 86, "...prompt diagnosis of ABF with infiltrative shadows and hematoma images seen in the lung fields around the aorta."  Line 88, "and hemoptysis, a cough that persisted for 3 months, and she ...".  Line 91, "...and the left lung and/or bronchus, leading to a...".  Line 120, "...blood and sputum cultures were negative."    

Author Response

Comments and Suggestions for Authors (Reviewer 1)

An interesting, educational and well written case report manuscript with an adequate review of the literature.  There some editing issues that the authors should consider and address.  The following are suggestions/comments regarding these editing issues.  Line 30, "...computed tomography (CT) in the outpatient department."  Lines 32 & 33, "On admission, a contrast-enhanced CT scan showed infiltrative....".  Line 69, "...after the surgery, a contrast-enhanced CT scan showed that the...".  Line 83, "...descending aorta, left lung, and/or bronchus [1,2]."  Lines 85 & 86, "...prompt diagnosis of ABF with infiltrative shadows and hematoma images seen in the lung fields around the aorta."  Line 88, "and hemoptysis, a cough that persisted for 3 months, and she ...".  Line 91, "...and the left lung and/or bronchus, leading to a...".  Line 120, "...blood and sputum cultures were negative."    

Response: Thank you for your suggestions, accordingly, we have revised our manuscript based on the revisions you suggested.

Reviewer 2 Report

The report should get some English proofreading.

Furthermore, some study has been reported about secondary aortobronchial fistula. The authors need to show new findings.

Comments

Please show the CT images at pre TEVAR with aortic dissection.  

Please show the change of the marker of infection more detailed.

Why did you select the LVFX for antibiotics?  

Author Response

Comments and Suggestions for Authors (Reviewer 2)

The report should get some English proofreading.

Furthermore, some study has been reported about secondary aortobronchial fistula. The authors need to show new findings.

Response: As you have pointed out, we recognize that the treatment and course of the disease in this case is similar to that in several previous case reports. However, we believe the lesson we intend to highlight in this case is that aortobronchial fistulas can still occur 18 years after graft replacement of the descending aorta. Another lesson is that long-term antibiotic therapy may not be necessary if culture results are negative.

Comments

Please show the CT images at pre TEVAR with aortic dissection.  

Response: Figure 1, C, shows residual dissection findings on the proximal side of the graft anastomosis. We have added that in the description of Figure 1 in response to your suggestion.

Please show the change of the marker of infection more detailed.

Response: The details of the postoperative infection process have been revised and added to lines 88 through 110 of the revised text.

Why did you select the LVFX for antibiotics?  

Response: Levofloxacin was selected because it is an antibiotic with a broad antibacterial spectrum that works against Gram-positive and Gram-negative bacteria and is also available orally.

Reviewer 3 Report

dear author, this is an interesting case description, however there are some issues that need clarification. 

Did you perform any other study? PET CT scan? In order to specify the location and expansion of the infection.

What type of imaging did you perform after the procedure?

Add in the abstract if the patient was on long-term antibiotics. 

Please expand your introduction which is currently short. There are many publications on this subject.

What was the indication for the initial operation?

Any blood cultures before intervention?

Why did you inserted first the larger stent graft? (28>26). It is more common to do the other way around for better overlap and oversizing. Additionally, the specific device is being deployed from the distal part which would be convenient in this case.

What type of iv antibiotics did you administer? What was the long term treatment? Any blood cultures during FU?

Those are some ref that you can add in the discussion:

Anastasiadou C, Trellopoulos G, Kastora S, Kakisis I, Papapetrou A, Galyfos G, Geroulakos G, Megalopoulos A. A systematic review of therapies for aortobronchial fistulae. J Vasc Surg. 2022 Feb;75(2):753-761.e3. doi: 10.1016/j.jvs.2021.08.108. Epub 2021 Oct 6. PMID: 34624495.

Yuan SM. Aortobronchial fistula. Gen Thorac Cardiovasc Surg. 2020 Feb;68(2):93-101. doi: 10.1007/s11748-019-01271-8. Epub 2020 Jan 1. PMID: 31894503.

Weaver ML, Black JH 3rd. Aortobronchial and aortoenteric fistula. Semin Vasc Surg. 2017 Jun-Sep;30(2-3):85-90. doi: 10.1053/j.semvascsurg.2017.10.005. Epub 2017 Oct 31. PMID: 29248125.

Author Response

Comments and Suggestions for Authors (Reviewer 3)

dear author, this is an interesting case description, however there are some issues that need clarification. 

Did you perform any other study? PET CT scan? In order to specify the location and expansion of the infection.

Response: Ga scintigraphy was performed 3 days before the surgery, and the results were added to Figure 1.

What type of imaging did you perform after the procedure?

Response: Postoperative imaging examinations included contrast-enhanced CT of the thorax and abdomen and chest radiography.

Add in the abstract if the patient was on long-term antibiotics.

Response: The only antibiotic administered in this case was oral levofloxacin for 14 days, and no other antibiotics were administered. Therefore, no long-term antibiotic therapy was used. The details of the postoperative infection process have been revised and added to lines 88 through 110 of the revised text.

Please expand your introduction which is currently short. There are many publications on this subject.

Response: We have added more information to the introduction according to your suggestion.

What was the indication for the initial operation?

Response: The first surgery, performed 18 years ago, was a descending thoracic aortic aneurysm, for which he underwent graft replacement of the descending aorta. This is noted on lines 33 and 34 of the revised text.

Any blood cultures before intervention?

Response: One blood culture was obtained before the surgery, but the result was negative. No other blood cultures were collected. The details of the postoperative infection process have been revised and added to lines 88 through 110 of the text.

Why did you inserted first the larger stent graft? (28>26). It is more common to do the other way around for better overlap and oversizing. Additionally, the specific device is being deployed from the distal part which would be convenient in this case.

Response: Reviewing the operative record, as usual, a 26-mm stent graft was placed on the distal site first, followed by a 28-mm stent graft on the proximal site. It was an oversight error, and we am very grateful for for pointing it out. We have also corrected the surgical findings in the revised text accordingly.

What type of iv antibiotics did you administer? What was the long term treatment? Any blood cultures during FU?

Response: As I mentioned earlier, the only antibiotic administered in this case was oral levofloxacin for 14 days, and no other antibiotics were administered. Therefore, no long-term antibiotic therapy was used. For blood cultures, one blood culture was performed before surgery, but the result was negative. No other blood cultures were performed. The details of the postoperative infection process have been revised and added to lines 88 through 110 of the revised text.

Those are some ref that you can add in the discussion:

Anastasiadou C, Trellopoulos G, Kastora S, Kakisis I, Papapetrou A, Galyfos G, Geroulakos G, Megalopoulos A. A systematic review of therapies for aortobronchial fistulae. J Vasc Surg. 2022 Feb;75(2):753-761.e3. doi: 10.1016/j.jvs.2021.08.108. Epub 2021 Oct 6. PMID: 34624495.

Yuan SM. Aortobronchial fistula. Gen Thorac Cardiovasc Surg. 2020 Feb;68(2):93-101. doi: 10.1007/s11748-019-01271-8. Epub 2020 Jan 1. PMID: 31894503.

Weaver ML, Black JH 3rd. Aortobronchial and aortoenteric fistula. Semin Vasc Surg. 2017 Jun-Sep;30(2-3):85-90. doi: 10.1053/j.semvascsurg.2017.10.005. Epub 2017 Oct 31. PMID: 29248125.

Response: Thank you for providing the references.

Round 2

Reviewer 2 Report

Please performed English proofreading.

Author Response

Comments and Suggestions for Authors (Reviewer 2)

Please performed English proofreading.

Response: We would like to thank the reviewer for evaluating our manuscript and for his/her comment. Please note that we have sent our manuscript to an English editing company (Editage) for English proofreading. We hope that the level of English has been significantly improved in the revised manuscript.

Reviewer 3 Report

dear authors, thank you for your response. I only suggest to add in the discussion that close FU is needed, especially in cases that not long-term antibiotics are prescribed. Potentially blood test and blood cultures may be needed and imaging with PET CT in some time of FU. Those patients are prone to re-infection especially in endovascular approach.

Author Response

Comments and Suggestions for Authors (Reviewer 3)

dear authors, thank you for your response. I only suggest to add in the discussion that close FU is needed, especially in cases that not long-term antibiotics are prescribed. Potentially blood test and blood cultures may be needed and imaging with PET CT in some time of FU. Those patients are prone to re-infection especially in endovascular approach.

Response: We would like to thank the reviewer for the insightful comment. To address the issue raised by the reviewer, we have added the following part to the end of the Discussion section:

“However, as the antibiotic administration duration was short in this case, there is a possibility of ABF recurrence or stent graft infection in the future. Therefore, we will continue to closely follow-up the patient with imaging studies, blood culture, and blood tests.”